# The Association between Loneliness and Health Related Quality of Life (HR-QoL) among Community-Dwelling Older Citizens

**DOI:** 10.3390/ijerph17020600

**Published:** 2020-01-17

**Authors:** Siok Swan Tan, Irene N. Fierloos, Xuxi Zhang, Elin Koppelaar, Tamara Alhambra-Borras, Tasos Rentoumis, Greg Williams, Tomislav Rukavina, Rob van Staveren, Jordi Garces-Ferrer, Carmen B. Franse, Hein Raat

**Affiliations:** 1Department of Public Health, Erasmus MC University Medical Center, P.O. Box 2040, 3000 CA Rotterdam, The Netherlands; i.fierloos@erasmusmc.nl (I.N.F.); x.zhang@erasmusmc.nl (X.Z.); c.franse@erasmusmc.nl (C.B.F.); h.raat@erasmusmc.nl (H.R.); 2Research Centre Innovation in Care, Rotterdam University of Applied Sciences, P.O. Box 25035, 3001 HA Rotterdam, The Netherlands; e.koppelaar@hr.nl; 3Polibienestar Research Institute, University of Valencia, Carrer del Serpis, 29, 46022 Valencia, Spain; tamara.alhambra@uv.es (T.A.-B.); jordi.garces@uv.es (J.G.-F.); 4Alliance for Integrated Care, Amfitritis 14, Palaio Faliro, 175 61 Athens, Greece; a.i.rentoumis@gmail.com; 5Division of Population Health, Health Services Research and Primary Care, University of Manchester, Oxford Road, Manchester M13 9PT, UK; greg.williams@manchester.ac.uk; 6Department of Social Medicine and Epidemiology, Faculty of Medicine, University of Rijeka, Trg braće Mažuranića 10, 51000 Rijeka, Croatia; tomislav.rukavina@medri.uniri.hr; 7Zorg Op Noord, Cypresbaan 7, 2908 LT Capelle aan den IJssel, The Netherlands; r.vanstaveren@zonboog.nl

**Keywords:** community-dwelling older citizens, emotional loneliness, health related quality of life, loneliness, social loneliness

## Abstract

*Background*: This study aimed to assess the association between loneliness and Health-Related Quality of Life (HR-QoL) among community-dwelling older citizens in five European countries. We characterize loneliness broadly from an emotional and social perspective. *Methods*: This cross-sectional study measured loneliness with the 6-item De Jong Gierveld Loneliness Scale and HR-QoL with the 12-Item Short-Form Health Survey. The association between loneliness and HR-QoL was examined using multivariable linear regression models. *Results*: Data of 2169 citizens of at least 70 years of age and living independently (mean age = 79.6 ± 5.6; 61% females) were analyzed. Among the participants, 1007 (46%) were lonely; 627 (29%) were emotionally and 575 (27%) socially lonely. Participants who were lonely experienced a lower HR-QoL than participants who were not lonely (*p* ≤ 0.001). Emotional loneliness [std-β: −1.39; 95%-CI: −1.88 to −0.91] and social loneliness [−0.95; −1.44 to −0.45] were both associated with a lower physical HR-QoL. Emotional loneliness [−3.73; −4.16 to −3.31] and social loneliness [−1.84; −2.27 to −1.41] were also both associated with a lower mental HR-QoL. *Conclusions*: We found a negative association between loneliness and HR-QoL, especially between emotional loneliness and mental HR-QoL. This finding indicates that older citizens who miss an intimate or intense emotional relationship and interventions targeting mental HR-QoL deserve more attention in policy and practice than in the past.

## 1. Introduction

Because of their decreasing social circle and increasing health restrictions, older citizens may be particularly vulnerable to feeling lonely [1,2,3]. Opposed to social isolation which is an objective measure of the number of social interactions that people have, loneliness is a subjective measure [4,5,6]. Loneliness has been described as the unpleasant subjective feeling that occurs when a person’s social environment is deficient in some important way, either quantitatively or qualitatively [3,6]. Hence, someone may be bodily around people but still feel lonely or be fully socially isolated but not feel lonely. Social isolation may evoke loneliness and vice versa; and both concepts can arise simultaneously [4,5,6]. From a policy perspective, alleviating loneliness cannot be achieved by solely increasing the number of social interactions, but by improving meaningful social networks [3,5].

Owing to socio-cultural differences, varying social care systems and the use of different instruments and definitions of loneliness, estimations on prevalence rates of loneliness vary extensively [7,8]. The European Social Survey (ESS) reported prevalence rates of frequent loneliness in the general population varying between 3% in the Netherlands to 10% in Greece [8]. A European cross-country comparison estimated the prevalence of loneliness in citizens of at least 60 years at below 6% in Northern Europe and up to 34% in Eastern Europe [7]. Owing to the ageing and increasing life expectancy of Western populations, these prevalence rates are expected to rapidly increase in the near future [7,8].

The association between loneliness and health is well-established. For example, loneliness was observed to be a risk factor for the development of depression, dementia, hypertension, cardiovascular disease and stroke in older citizens. Conversely, loneliness can be an outcome of health when poor health conditions lead to physical and mental restrictions [9,10]. Therefore, the number and variety of interventions which target loneliness, such as social facilitation, psychological therapies, leisure/skill development and smart technology, have increased in recent years [11,12,13]. However, despite efforts to develop evidence-based interventions, few interventions have been effective in alleviating loneliness [14,15]. Characteristics of interventions which show a positive impact on loneliness include a) the adaptability of an intervention to a local context and b) the focus of an intervention on active engagement [14,15].

Health-Related Quality of Life (HR-QoL) is a bi-dimensional construct focusing on health-related aspects of well-being and reflecting the subjective perception of the impact of physical and mental functioning on a person’s daily living [16]. A recent study of Gerino et al. (2017) identified two mechanisms of the impact of loneliness on HR-QoL among older citizens [17]. The first considers that older citizens who are lonely are at increased risk of poor health, resulting in low HR-QoL. The second considers that older citizens who are lonely may experience a lower level of resilience, resulting in low HR-QoL [17,18]. Previous studies examining the association between loneliness and HR-QoL indeed consistently reported loneliness to be an important determinant for low HR-QoL [1,2,17,19,20,21]. However, the studies share two important features: a) they were carried out in single countries, which restricts the generalizability of their results to other local contexts and b) they predominantly focused on social loneliness. Weiss and colleagues distinguished two dimensions of loneliness in 1973-emotional and social loneliness [22]. Emotional loneliness concerns the feeling of missing an intimate or intense emotional relationship (e.g., partner, relative or close friend), whereas social loneliness concerns the feeling of missing social interactions (e.g., with friends, co-workers or neighbors) [22,23]. Exploring both dimensions of loneliness separately could provide professionals with valuable insight to establish tailored support to actively engage lonely citizens at risk for suboptimal HR-QoL.

The aim of the present study is to examine the association between loneliness and HR-QoL among community-dwelling older citizens in five European countries. We characterize loneliness broadly from an emotional and social perspective. We hypothesize that overall loneliness is associated with lower physical and mental HR-QoL. Also, we hypothesize that emotional loneliness has a stronger association with HR-QoL than social loneliness and that loneliness has a stronger association with mental HR-QoL than with physical HR-QoL.

## 2. Materials and Methods

The Urban Health Centers Europe (UHCE) project aimed to improve the management of multi-morbidity of older citizens using integrated care pathways that focus on adherence to treatment and prevention of falls and frailty (https://www.age-platform.eu/project/urban-health-centres-europe-uhce). The study design has been described in detail elsewhere [24,25]. In short, integrated care pathways were implemented in primary care and community settings in five European countries (the United Kingdom, Greece, Croatia, the Netherlands and Spain). Citizens were eligible to enroll when they were at least 75 years of age and lived independently. In Greece and Spain, citizens of at least 70 were also eligible owing to difficulty in inclusion. Citizens were not eligible when they lacked the basic knowledge of the local language or when they were not expected to be able to make an informed decision regarding participation—According to their physician. A total of 2325 participants were recruited between May 2015 and June 2017, of which 1215 enrolled to receive integrated care pathways (intervention) and 1110 enrolled in the control group. Each participant completed a self-reported questionnaire in their local language at baseline and at 12-months follow-up. Informed consent was obtained from each participant and ethical approval was received from the medical ethics committees in each country.

For the current cross-sectional study, baseline data of the UHCE project were used (*n* = 2325). Participants with missing data on overall, emotional and/or social loneliness (*n* = 35), HR-QoL (*n* = 117) and on age (*n* = 1) or sex (*n* = 3) were excluded. Hence, the current study included 2169 participants, with a mean age of 79.6 ± 5.6 years and 61% females.

The questionnaire included the 6-item De Jong Gierveld Loneliness scale and the 12-Item Short-Form Health Survey (SF-12) version 2. The De Jong Gierveld Loneliness scale is a reliable and valid instrument to assess overall loneliness in adults of all ages [23]. The scale comprises the dimensions of emotional and social loneliness. Emotional loneliness describes the general sense of emptiness, the feeling of not having people around and rejection (three items). Items have answer categories of no (score 0), more or less (1) and yes (1). Social loneliness describes the feeling of having people to rely on, the feeling of having people to trust completely and the feeling of having people to feel close to (three items). Items have answer categories of no (score 1), more or less (1) and yes (0). Thus, the dimension scores range from 0–3 and the overall loneliness score from 0–6, with higher scores indicating a higher experience of loneliness. Participants were considered to be lonely (score ≥ 2) or not lonely (score < 2). Owing to the cut-off point at score 2 for both overall loneliness (6 items) and its two dimensions (three items each), the presence of overall loneliness is not deductible from those of emotional and social loneliness and vice versa. For example, a citizen who gets a score indicating overall loneliness (score = 2) may not be emotionally or socially lonely (if both scores equal 1).

The SF-12 version 2 measures self-perceived burden of illness and has shown to be a reliable and valid standardized instrument to assess HR-QoL in the adult community- and chronic disease populations [26]. The questionnaire comprises of eight domains: bodily pain (1 item), vitality (1), physical functioning (2), physical role functioning (2), emotional role functioning (2), mental health (2), social functioning (1) and general health (1). SF-12 scores can be summarized into the 12-item Physical Component Summary (PCS-12) and the 12-item Mental Component Summary (MCS-12), with higher scores indicating higher levels of health.

Socio-demographic characteristics were assessed and incorporated as covariates: age, sex, country, living situation, level of education, alcohol risk, physical activity, smoking and multi-morbidity. With respect to living situation, participants were categorized into not living with others or living with others (partner, child (ren) and/or others). The level of education concerned the highest level the participant ever completed and was categorized according to the 2011 International Standard Classification of Education (ISCED) into primary or less (ISCED level 0–1), secondary or equivalent (2–5) and tertiary or higher (6–8). Alcohol risk was assessed with three items of the AUDIT-C on high-risk alcohol use. AUDIT-C scores range from 0–12, with higher scores indicating a greater likelihood that drinking is affecting the participant’s health and safety. Participants were categorized into no alcohol risk (score < 4 for men; score < 3 for women) or alcohol risk (score ≥ 4 for men; score ≥ 3 for women) [27]. Physical activity concerned the frequency a participant engaged in activities that require low or moderate energy (once a week or less; more than once a week). Finally, multi-morbidity was defined as having at least two of the following chronic conditions: heart attack, hypertension, diabetes, stroke, high blood cholesterol, asthma, arthritis, osteoporosis, chronic lung disease, cancer or malignant tumor, stomach or duodenal ulcer, Parkinson’s disease, cataract and hip or femoral fracture [28].

Descriptive statistics of the covariates were determined and stratified by dimension of loneliness. Differences in HR-QoL scores were compared between participants who were and were not lonely. Cohen’s effect sizes (d) were used for interpretation of relevant differences: 0.20 ≤ d <0.50 was considered a small, 0.50 ≤ d < 0.80 a moderate and d ≥ 0.80 a large difference [29]. The association between loneliness and HR-QoL was examined using multivariable linear regression models. The crude models only included loneliness (yes vs no), the adjusted models additionally incorporated the covariates and the full models included both dimensions of loneliness and the covariates. We calculated standardized (std)-β’s to be able to compare the relative importance of each β in the regression models. The 95%-confidence intervals (CIs) of the estimated β’s indicated whether β’s were significantly different. We performed tests for linearity, tests for normality of residuals with kernel density plots and tests for multicollinearity with variance inflation factors. No violation of basic assumptions for regression and no multicollinearity problems were found. Finally, Univariate Analysis of Variance (UNIANOVA) was applied to assess interactions between emotional and/or social loneliness and country as well as between emotional and/or social loneliness and other covariates in the association between (emotional and/or social) loneliness and HR-QoL. After applying Bonferroni correction for multiple testing, the *p*-value of the interaction analyses equaled *p* = 0.05/30 = 0.002. In all other cases, *p* < 0.05 was taken as statistically significant. Analysis were performed with SPSS version 25.0 (IBM SPSS Statistics for Windows, IBM Corp.: Armonk, NY, USA).

## 3. Results

Table 1 presents the general characteristics of the study population. About half of the participants were lonely (46%). The prevalence rate of loneliness varied between 31% in the United Kingdom and 74% in Croatia (*p* < 0.001). Compared with participants who were not lonely, participants who were lonely were older (*p* < 0.001), more often female (*p* < 0.001), more often completed a lower level of education (*p* = 0.012), more often lived alone (*p* < 0.001), were less often at risk for alcohol use (*p* = 0.029), less often engaged in physical activity more than once a week (*p* < 0.001) and more often suffered from multi-morbidity (*p* = 0.002).

Table 2 presents the number of participants who were emotionally and socially lonely. About half of the participants were emotionally and/or socially lonely (42%). Of those, 291 were both emotionally and socially lonely (32%), 336 were only emotionally lonely (37%) and 284 were only socially lonely (31%). Appendix A summarizes the general characteristics of participants stratified by dimension of loneliness.

The HR-QoL of the total population was comparable to that of Dutch reference values in the age group 70–79 years: the PCS-12 score was 41.8 ± 12.1 (versus 44.2 ± 11.7) and the MCS-12 score 50.2 ± 10.7 (versus 47.7 ± 10.5) [30]. Table 3 presents the correlation between emotional and/or social loneliness and HR-QoL. Compared with participants who were not lonely, participants who were emotionally and/or socially lonely had a lower HR-QoL (*p* < 0.001). For mental HR-QoL, the differences between participants who were and were not emotionally and/or socially lonely was large (d ≥ 0.80).

Table 4 shows the association between emotional and/or social loneliness and physical HR-QoL and Appendix A the std-β’s of the full model and their 95%-CIs. 

Table 5 shows the association between emotional and/or social loneliness and mental HR-QoL. Loneliness was associated with lower mental HR-QoL (*p* < 0.001). In the crude model, older citizens who are lonely score 9.73 points less on the MCS-12 than those who are not lonely. Adjusted for the co-variates, older citizens who are lonely score 7.64 points less on the MCS-12. Being female, living in the UK, Greece or Croatia, a low level of education, being at risk for alcohol use and a low level of physical activity significantly affected mental HR-QoL. Emotional loneliness [β: −8.19; std-β: −3.73; 95%-CI: −4.16 to −3.31] had a stronger association with mental HR-QoL than social loneliness [β: −4.16; std-β: −1.84; 95%-CI: −2.27 to −1.41]. Similar results were observed for the dimensions of loneliness separately. The amount of variance explained by the emotional loneliness model was higher than that by the social loneliness model (32% vs. 24%). Emotional loneliness also had a stronger association with mental HR-QoL [std-β: −3.73; 95%-CI: −4.16 to −3.31] than with physical HR-QoL [std-β: −1.39; 95%-CI: −1.88 to −0.91]. Comparing the association between social loneliness and mental HR-QoL [std-β: −1.84; 95%-CI: −2.27 to −1.41] with that between social loneliness and physical HR-QoL [std-β: −0.95; 95%-CI: −1.44 to −0.45] reached borderline significance.

Appendix A presents the *p*-values of the interaction analyses. A statistically significant interaction between emotional loneliness and country was found. Appendix A shows the association between emotional loneliness and HR-QoL stratified by country. With the exception of Greece, emotional loneliness was significantly associated with physical HR-QoL. Emotional loneliness was significantly associated with mental HR-QoL in all countries. The explained variance of physical HR-QoL ranged from 19% in Spain to 35% in Croatia and that of mental HR-QoL from 17% in the UK to 36% in Croatia.

## 4. Discussion

Our study aimed to examine the association between loneliness and HR-QoL among community-dwelling older citizens in five European countries. In agreement with the existing literature [1,2,17,19,20,21], we found that citizens who were lonely experience a lower physical and mental HR-QoL than citizens who were not lonely. The difference between lonely and not lonely citizens was especially large for mental HR-Qo, confirming the strong association between loneliness and mental HR-QoL.

Using the De Jong Gierveld Loneliness scale allowed us to address emotional and social loneliness each separately. The association between emotional loneliness and physical HR-QoL was as strong as the association between social loneliness and physical HR-QoL. However, emotional loneliness has a stronger association with mental HR-QoL than social loneliness and loneliness has a stronger association with mental HR-QoL than with physical HR-QoL. The latter finding was also observed in earlier studies using the UCLA Loneliness scale and the PCS and MCS dimensions of the SF-12/36 to assess the association between loneliness and HR-QoL [1,20]. The UCLA Loneliness scale measures loneliness as a one-dimensional concept in adults of all ages and especially covers the dimension of social loneliness [23,31]. In addition, living alone was a risk factor for mental (and not physical) HR-QoL only in addition to emotional (and not social) loneliness. These results may imply that citizens who miss an intimate or intense emotional relationship and live alone are at particular risk for a lower mental HR-QoL and confirms that both concepts can arise separately and simultaneously. Earlier studies have already suggested that having a partner and/or having (adult) children relate positively to well-being and mental health [32,33]. Our findings underline the need to address the two loneliness dimensions separately.

Our study was conducted in five European countries. We found significant differences between countries. In agreement to earlier studies [7,8], we found that the loneliness rate in Croatia (74%) and Spain (55%) was higher than in the other countries (35% on average). Furthermore, older citizens living in the United Kingdom, Croatia and the Netherlands scored about 4.5 points less on the PCS-12 than those living in Greece and Spain. Older citizens living in Croatia scored about 5 points less and older citizens living in the United Kingdom about 1.5 points less on the MCS-12 than those living in and Spain. Also, there was a significant interaction between ‘emotional loneliness’ and ‘country’. Quality of living conditions, level of social integration, individual social expectations, strength of societal welfare, demographic composition and cultural norms and values have been suggested to explain these cross-country differences [6,7]. Future studies should clarify the differences observed, particularly because earlier studies report contradicting conclusions on the positive or negative association between loneliness and country-specific characteristics [34,35].

Our study has some limitations which should be considered in interpreting the results. Firstly, the cross-sectional design of our study did not allow to establish the causal (uni- or bi-directional) relationship between loneliness and HR-QoL. However, our findings support the need for further research on evaluating the effects of loneliness as well as the two dimensions of loneliness on HR-QoL.

Secondly, our regression models adjusted for potential covariates. Covariates were restricted by data availability. In addition, some of the covariates may act as partially a mediator. For example, poor health may jeopardize the social and physical activity of a citizen which could nourish loneliness. Or: being lonely might mean that a citizen does not have a reason to go out which results in lower physical activity. This sub-optimal physical activity consequently effects the (physical) HR-QoL. In that case, the addition of physical activity may have resulted in an underestimation of HR-QoL (overfitting).

Finally, we did not assess the socio-cultural differences in the interpretation of individual items between countries. Consequently, we may have observed some unintended variation between countries. Still, we have paid specific attention to translating the items of the De Jong Gierveld Loneliness scale for which no validated translation was available (Croatia, Greece). Items were translated forward and backward and the translations were discussed by the study team, adapted when needed and piloted in at least five older citizens.

## 5. Conclusions

To our knowledge, this is the first study to explore the association between the two dimensions of loneliness and HR-QoL. We used rich data of a large sample of community-dwelling older citizens in five European countries contributing to the generalizability of the results to other local contexts. Exploring both dimensions of loneliness separately provides professionals with valuable insight to establish tailored support to actively engage lonely citizens at risk for suboptimal HR-QoL. Our study showed a negative association between loneliness and HR-QoL, especially between emotional loneliness and mental HR-QoL. This finding suggest that older citizens who miss an intimate or intense emotional relationship and interventions targeting mental HR-QoL deserve more attention in policy and practice than in the past. Future studies should examine socio-demographic characteristics associated with emotional and social loneliness and identify characteristics of interventions for specific risk populations to more effectively alleviate their loneliness. Policy makers should ensure that social care systems accommodate the physical and social environments required to support lonely citizens at risk for suboptimal HR-QoL.

## Figures and Tables

**Table 1 ijerph-17-00600-t001:** General characteristics of the study population.

Items	Total (*n* = 2169) Mean ± SD N (%)	Loneliness		
		Yes (*n* = 1007) Mean ± SD N (%)	No (*n* = 1162) Mean ± SD N (%)	*p*-Value
Age	79.6 ± 5.6	80.2 ± 5.6	79.1 ± 5.5	<0.001 ***
Sex				<0.001 ***
Female	1313 (60.5)	655 (49.9)	658 (50.1)	
Male	856 (39.5)	352 (41.1)	504 (58.9)	
Country				<0.001 ***
The United Kingdom	528 (24.3)	162 (30.7)	366 (69.3)	
Greece	331 (15.3)	182 (55.0)	149 (45.0)	
Croatia	481 (22.2)	356 (74.0)	125 (26.0)	
The Netherlands	336 (15.5)	126 (37.5)	210 (62.5)	
Spain	493 (22.7)	181 (36.7)	312 (63.3)	
Level of education +				0.012 *
Primary or less	588 (27.4)	283 (48.1)	305 (51.9)	
Secondary or equivalent	1362 (63.6)	639 (46.9)	723 (53.1)	
Tertiary or higher	193 (9.0)	70 (36.3)	123 (63.7)	
Living situation +				<0.001 ***
Living with others	1340 (62.0)	555 (41.4)	785 (58.6)	
Living alone	823 (38.0)	448 (54.4)	375 (45.6)	
Alcohol risk +				0.029 *
No	1522 (73.7)	722 (47.4)	800 (52.6)	
Yes	543 (26.3)	228 (42.0)	315 (58.0)	
Physical activity +				<0.001 ***
<once a week	610 (28.3)	381 (62.5)	229 (37.5)	
≥once a week	1545 (71.7)	616 (39.9)	929 (60.1)	
Smoking +				0.857
No	2007 (92.7)	931 (46.4)	1076 (53.6)	
Yes	157 (7.3)	74 (47.1)	83 (52.9)	
Multi-morbidity +				0.002 **
No	197 (9.1)	71 (36.0)	126 (64.0)	
Yes	1970 (90.9)	934 (47.4)	1036 (52.6)	

Notes: SD = standard deviation; + Missing items: Level of education = 26; Living situation = 6; Alcohol risk = 104; Physical activity = 14; Smoking = 5; Multi-morbidity = 2; * *p* < 0.05, ** *p* < 0.01, *** *p* < 0.001, *p*-values are based on Independent T test for participants who are not lonely and participants who are lonely.

**Table 2 ijerph-17-00600-t002:** Number of participants experiencing emotional and/or social loneliness.

		Social Loneliness		Total
		Yes	No	
Emotional loneliness	Yes	291 (14%)	336 (15%)	627
	No	284 (13%)	1258 (58%)	1542
Total		575	1594	2169

**Table 3 ijerph-17-00600-t003:** Health-Related Quality of Life of the study population stratified by loneliness.

Items	Health-Related Quality of Life Mean ± SD	
	PCS-12	MCS-12
Total (*n* = 2169)	41.78 ± 12.06	50.18 ± 10.73
Loneliness		
Yes (*n* = 1007)	38.53 ± 12.23	44.98 ± 10.89
No (*n* = 1162)	44.60 ± 11.19	54.71 ± 8.29
Effect Size ^+^	0.52 ***	1.01 ***
Emotional loneliness		
Yes (*n* = 627)	37.53 ± 12.12	42.45 ± 10.66
No (*n* = 1542)	43.51 ± 11.61	53.34 ± 9.05
Effect Size +	0.50 ***	0.83 ***
Social loneliness		
Yes (*n* = 575)	37.82 ± 11.80	43.83 ± 11.48
No (*n* = 1594)	43.21 ± 11.84	52.49 ± 9.47
Effect Size +	0.46 ***	0.82 ***

Notes: PCS-12 = 12-item Physical Component Summary, with higher scores indicating higher levels of health; MCS-12 = 12-item Mental Component Summary, with higher scores indicating higher levels of health. SD = standard deviation; + Cohen’s effect size (d) for differences in HRQOL between participants who are lonely and participants are not lonely. 0.20 ≤ d < 0.50 is considered a small difference; 0.50 ≤ d < 0.80 a moderate difference; d ≥ 0.80 a large difference. *** *p* ≤ 0.001, *p*-values are based on Independent T test for participants who are not lonely and participants who are lonely.

**Table 4 ijerph-17-00600-t004:** Association between loneliness and physical Health-Related Quality of Life.

	PCS-12						
	Overall Loneliness		Emotional Loneliness		Social Loneliness		Emotional and Social Loneliness
	Crude Model	Adjusted Model	Crude Model	Adjusted Model	Crude Model	Adjusted Model	Full Model
Loneliness (*Yes versus No*)	−6.52 ***	−3.71 ***					
Emotional loneliness			−6.53 ***	−3.53 ***			−3.06 ***
Social loneliness					−5.95 ***	−2.86 ***	−2.14 ***
Age		−0.26 ***		−0.27 ***		−0.27 ***	−0.26 ***
Sex							
Female		−2.58 ***		−2.45 ***		−2.70 ***	−2.52 ***
Male (*ref*)							
Country							
The United Kingdom		−4.90		−4.80 ***		−4.77 ***	−2.14 ***
Greece		−0.92		−0.82		−1.54	−0.87
Croatia		−4.59 ***		−5.30 ***		−4.88 ***	−4.59 ***
The Netherlands		−4.25 ***		−4.21 ***		−4.20 ***	−4.14 ***
Spain (*ref*)							
Level of education							
Primary or less		−2.29 *		−2.33 **		−2.51 **	−2.23 *
Secondary or equivalent		0.23		0.17		0.16	0.19
Tertiary or higher (*ref*)							
Living situation							
Living alone		0.52		0.55		0.15	0.54
Living with others (*ref*)							
Alcohol risk							
No (*ref*)							
Yes		2.10 ***		1.97 ***		1.96 ***	1.96 ***
Physical activity							
< once a week		−9.04 ***		−9.10 ***		−9.27 ***	−8.91 ***
≥ once a week (*ref*)							
Smoking							
No (*ref*)							
Yes		0.40		0.49		0.47	0.51
Multi-morbidity							
No (*ref*)							
Yes		−6.19 ***		−6.09 ***		−6.35 ***	−6.06 ***
Adjusted R square, %	7.2	31.4	5.9	31.0	4.7	30.4	31.5

Notes: PCS-12 = 12-item Physical Component Summary, with higher scores indicating higher levels of health. ref = reference category; * *p* ≤ 0.05, ** *p* ≤ 0.01, *** *p* ≤ 0.001, *p*-values are based on multivariable linear regression models.

**Table 5 ijerph-17-00600-t005:** Association between loneliness and mental Health-Related Quality of Life.

	MCS-12						
	Overall Loneliness		Emotional Loneliness		Social Loneliness		Emotional and Social loneliness
	Crude Model	Adjusted Model	Crude Model	Adjusted Model	Crude Model	Adjusted Model	Full Model
Loneliness (*Yes versus No*)	−9.73 ***	−7.64 ***					
Emotional loneliness			−10.95 ***	−9.10 ***			−8.19 ***
Social loneliness					−8.71 ***	−6.10 ***	−4.16 ***
Age		0.08		0.06		0.05	0.07
Sex							
Female		−2.01 ***		−1.65 ***		−2.27 ***	−1.78 ***
Male (*ref*)							
Country							
The United Kingdom		−1.79 *		−1.66 **		−1.54 *	−1.79 **
Greece		−1.71 *		−1.08		−2.97 ***	−1.17
Croatia		−4.90 ***		−6.02 ***		−5.42 ***	−4.65 ***
The Netherlands		1.18		1.33		1.28	1.47 *
Spain (*ref*)							
Level of education							
Primary or less		−2.26 **		−2.15 **		−2.69 **	−1.94 **
Secondary or equivalent		−1.15		−1.24		−1.30	−1.20
Tertiary or higher (*ref*)							
Living situation							
Living alone		0.87		1.19 **		0.12	1.17 **
Living with others (*ref*)							
Alcohol risk							
No (*ref*)							
Yes		1.38 **		1.12 *		1.11 *	1.10 **
Physical activity							
< once a week		−4.40 ***		−4.26 ***		−4.85 ***	−3.88 ***
≥ once a week (*ref*)							
Smoking							
No (*ref*)							
Yes		−0.61		−0.40		−0.47	−0.37
Multi-morbidity							
No (*ref*)							
Yes		−0.90		−0.51		−1.22	−0.46
Adjusted R square, %	20.3	29.9	21.0	32.1	12.7	24.1	34.4

Notes: MCS-12 = 12-item Mental Component Summary, with higher scores indicating higher levels of health; ref = reference category; * *p* ≤ 0.05, ** *p* ≤ 0.01, *** *p* ≤ 0.001, *p*-values are based on multivariable linear regression models.

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
