# Peer review of "The Association between Loneliness and Health Related Quality of Life (HR-QoL) among Community-Dwelling Older Citizens"

_ijerph, 2020, doi:10.3390/ijerph17020600_

Round 1

Reviewer 1 Report

This study examined the relationship of loneliness with health-related quality of life among community-based older adults in five European countries. The impact of loneliness on HRQOL has emerging to be an important issue for older adults, and the data from multiple European countries was valuable. Here are some comments or suggestions for the authors.

Definition and measures of loneliness: The definition of loneliness should be stated in the Introduction, and the difference of loneliness and social isolation may be discussed. Also the measure of loneliness included two dimensions of emotional loneliness and social loneliness by the DeJong Gierveld Loneliness Scale. Whether the difference of social isolation and social loneliness are different or the same could be discussed. In the literature review, the effect of loneliness to health focused on diseases, the interventions, and effects on HRQOL were introduced by empirical studies. The authors seem to view loneliness as a disease or a risk factor for health, but loneliness can also be an outcome of health conditions or social issues. The explanations or possible mechanism about loneliness on physical and mental health, or on quality of life are needed. The loneliness rate in Croatia was much higher (about double) than other countries. Also in Table S1, social loneliess of Croatia was much higher. Was there any reason that loneliness of Croatia elderly was so high? Please explain. In addition, the authors emphasize one of the strengths of this study was the data were from multiple countries. The background of the five countries should also be provided in the Introduction section, and the country differences should be discussed in the Discussion section. Other related factors: There are possible covariates related to loneliness and HRQOL can be considered, such as marital status, living arrangement, social support, social participation, etc. If some of the related factors are available, they can be controlled in the model. The significance symbol in Tables can be replace by star (*p<0.05, **p<0.01, ***p<0.001). The implications based on the results of associations between two dimensions of loneliness with HRQOL should be elaborated more, and the future research suggestions can be added.

Author Response

Reviewer 1

This study examined the relationship of loneliness with health-related quality of life among community-based older adults in five European countries. The impact of loneliness on HRQOL has emerging to be an important issue for older adults, and the data from multiple European countries was valuable. Here are some comments or suggestions for the authors. Definition and measures of loneliness: The definition of loneliness should be stated in the Introduction, and the difference of loneliness and social isolation may be discussed. Also the measure of loneliness included two dimensions of emotional loneliness and social loneliness by the DeJong Gierveld Loneliness Scale. Whether the difference of social isolation and social loneliness are different or the same could be discussed.

In line with the reviewers suggestion, we now start the introduction with the definition of loneliness, as well as an elaboration on the difference between loneliness and social isolation: ‘Because of their decreasing social circle and increasing health restrictions, older citizens may be particularly vulnerable to feeling lonely [1-3]. Opposed to social isolation which is an objective measure of the number of social interactions that people have, loneliness is a subjective measure [4-6]. Loneliness has been described as the unpleasant subjective feeling that occurs when a person’s  social environment is deficient in some important way, either quantitatively or qualitatively [3, 6]. Hence, someone may be bodily around people but still feel lonely or be fully socially isolated but not feel lonely. Social isolation may evoke loneliness and vice versa; and both concepts can arise simultaneously [4-6]. From a policy perspective, alleviating loneliness cannot be achieved by solely increasing the number of social interactions, but by improving meaningful social networks [3, 5].’

In the literature review, the effect of loneliness to health focused on diseases, the interventions, and effects on HRQOL were introduced by empirical studies. The authors seem to view loneliness as a disease or a risk factor for health, but loneliness can also be an outcome of health conditions or social issues.

We agree with the reviewer that loneliness can also be an outcome of health conditions or social issues; the association comes both ways. We have adapted this in the introduction, as follows: ‘The association between loneliness and health is well-established. For example, loneliness was observed to be a risk factor for the development of depression, dementia, hypertension, cardiovascular disease and stroke in older citizens. Conversely, loneliness can be an outcome of health when poor health conditions lead to physical and emotional restrictions [9, 10].’

The explanations or possible mechanism about loneliness on physical and mental health, or on quality of life are needed.

In line with the reviewers comment, we elaborate on mechanisms about loneliness on physical and mental health and on quality of life in the introduction: ‘A recent study of Gerino et al. (2017) identified two mechanisms of the impact of loneliness on HR-QoL among older citizens [17]. The first considers that older citizens who are lonely are at increased risk of poor health, resulting in low HR-QoL. The second considers that older citizens who are lonely may experience a lower level of resilience, resulting in low HR-QoL [17, 18].’

The loneliness rate in Croatia was much higher (about double) than other countries. Also in Table S1, social loneliness of Croatia was much higher. Was there any reason that loneliness of Croatia elderly was so high? Please explain.

There are several European cross-country studies on the prevalence of loneliness and each of them suggest that loneliness rates are generally much higher in Eastern Europe (e.g. Croatia, Greece) as compared to Northern and Western Europe (e.g. the UK, the Netherlands and Spain). In the introduction we present this as follows: ‘The European Social Survey (ESS) reported prevalence rates of frequent loneliness in the general population varying between 3% in the Netherlands to 10% in Greece [8]. A European cross-country comparison estimated the prevalence of loneliness in citizens of at least 60 years at below 6% in Northern Europe and up to 34% in Eastern Europe [7].’

In the discussion-section we further explain: ‘We found significant differences between countries. In agreement to earlier studies [7, 8], we found that the loneliness rate in Croatia (74%) and Spain (55%) was higher than in the other countries (35% on average). […] Quality of living conditions, level of social integration, individual social expectations, strength of societal welfare, demographic composition and cultural norms and values have been suggested to explain these cross-country differences [6, 7]. Future studies should clarify the differences observed, particularly because earlier studies report contradicting conclusions on the positive or negative association between loneliness and country-specific characteristics [34, 35].’

In addition, the authors emphasize one of the strengths of this study was the data were from multiple countries. The background of the five countries should also be provided in the Introduction section, and the country differences should be discussed in the Discussion section.

We consciously aimed to avoid making the current study a cross-country comparison, as that would require much more context and background information on each of the countries – which would distract us from answering our primary objective (to examine the association between loneliness and HR-QoL among community-dwelling older citizens). We therefore considered the wording of the title, objective and methods very carefully. The result is that we consistently state to have used a sample of older citizens in five European countries – avoiding the need of explaining country differences. However, we do agree with the reviewer that such a cross-country comparison would be very valuable. We therefore have expanded the paragraph on cross-country differences in the discussion-section, as follows: ‘Our study was conducted in five European countries. We found significant differences between countries. In agreement to earlier studies [7, 8], we found that the loneliness rate in Croatia (74%) and Spain (55%) was higher than in the other countries (35% on average). Furthermore, older citizens living in the United Kingdom, Croatia and the Netherlands scored about 4.5 points less on the PCS-12 than those living in Greece and Spain. Older citizens living in Croatia scored about 5 points less and older citizens living in the United Kingdom about 1.5 points less on the MCS-12 than those living in and Spain. Also, there was a significant interaction between ‘emotional loneliness’ and ‘country’. Quality of living conditions, level of social integration, individual social expectations, strength of societal welfare, demographic composition and cultural norms and values have been suggested to explain these cross-country differences [6, 7]. Future studies should clarify the differences observed, particularly because earlier studies report contradicting conclusions on the positive or negative association between loneliness and country-specific characteristics [34, 35].’

Other related factors: There are possible covariates related to loneliness and HRQOL can be considered, such as marital status, living arrangement, social support, social participation, etc. If some of the related factors are available, they can be controlled in the model.

Our study uses data of the Urban Health Centers Europe (UHCE) project. The dataset included a restricted number of covariates, which we have included into the model (age, sex, country, living situation, level of education, alcohol risk, physical activity, smoking and multi-morbidity). We state in the discussion-section: ‘Secondly, our regression models adjusted for potential covariates. Covariates were restricted by data availability.’

The significance symbol in Tables can be replace by star (*p<0.05, **p<0.01, ***p<0.001).

We replaced the significance symbols in all the tables as well as supplements by star (*p<0.05, **p<0.01, ***p<0.001).  

The implications based on the results of associations between two dimensions of loneliness with HRQOL should be elaborated more, and the future research suggestions can be added.

Our study has a very clear conclusion, which we present as follows:

‘These results may imply that citizens who miss an intimate or intense emotional relationship are at particular risk for a lower mental HR-QoL.’

In addition, we elaborate a bit more on the results of associations between two dimensions of loneliness with HR-QoL, as follows:

‘Using the De Jong Gierveld Loneliness scale [23] allowed us to address emotional and social loneliness each separately. The association between emotional loneliness and physical HR-QoL was as strong as the association between social loneliness and physical HR-QoL. However, emotional loneliness has a stronger association with mental HR-QoL than social loneliness and loneliness has a stronger association with mental HR-QoL than with physical HR-QoL. The latter finding was also observed in earlier studies using the UCLA Loneliness scale and the PCS and MCS dimensions of the SF-12/36 to assess the association between loneliness and HR-QoL [1, 20]. The UCLA Loneliness scale measures loneliness as a one-dimensional concept in adults of all ages and especially covers the dimension of social loneliness [23, 31]. In addition, living alone was a risk factor for mental (and not physical) HR-QoL only in addition to emotional (and not social) loneliness. These results may imply that citizens who miss an intimate or intense emotional relationship and live alone are at particular risk for a lower mental HR-QoL and confirms that both concepts can arise separately and simultaneously. Earlier studies have already suggested that having a partner and/or having (adult) children relate positively to well-being and mental health [32, 33]. Our findings underline the need to address the two loneliness dimensions separately.’

Future research suggestions were added:

‘Future studies should clarify the differences observed, particularly because earlier studies report contradicting conclusions on the positive or negative association between loneliness and country-specific characteristics [34, 35].’

‘Future studies should examine socio-demographic characteristics associated with emotional and social loneliness and identify characteristics of interventions for specific risk populations to more effectively alleviate their loneliness.’

Reviewer 2 Report

I would like to commend the authors on the high quality research paper submitted to the journal. My main concern is the age criterion that the authors used to include participants in the study. Can the authors provide a strong rationale behind including patients only 70 yrs & above in the study. The other main comment I have is for the authors to interpret the coefficients of the linear regression rather than just presenting the results. Please see my comments attached.

Good luck!

Author Response

Reviewer 2

I would like to commend the authors on the high quality research paper submitted to the journal. My main concern is the age criterion that the authors used to include participants in the study. Can the authors provide a strong rationale behind including patients only 70 yrs & above in the study.

We thank the reviewer for his/her compliment.

Our study uses data of the Urban Health Centers Europe (UHCE) project, which primary aim was to improve the management of multi-morbidity of older citizens of at least 70  (https://www.age-platform.eu/project/urban-health-centres-europe-uhce). This dataset proved to be an excellent basis to define secondary objectives – one of which was addressed in the current paper. Because of their decreasing social circle and increasing health restrictions, older citizens of at least 70 may be particularly vulnerable to feeling lonely [1-3]. Therefore, we feel that assessing citizens ‘only’ 70 years & above to answer our study objective (to examine the association between loneliness and HR-QoL among community-dwelling older citizens in five European countries.) is justified.

The other main comment I have is for the authors to interpret the coefficients of the linear regression rather than just presenting the results.

We now interpret the coefficients of the linear regression more extensively: ‘Loneliness was associated with lower physical HR-QoL (P<0.001). In the crude model, older citizens who are lonely score 6.52 points less on the PCS-12 than those who are not lonely. Adjusted for the co-variates, older citizens who are lonely score 3.71 points less on the PCS-12.’

And: ‘Table 5 shows the association between emotional and/or social loneliness and mental HR-QoL. Loneliness was associated with lower mental HR-QoL (P<0.001). In the crude model, older citizens who are lonely score 9.73 points less on the MCS-12 than those who are not lonely. Adjusted for the co-variates, older citizens who are lonely score 7.64 points less on the MCS-12. Being female, living in the UK, Greece or Croatia, a low level of education, being at risk for alcohol use and a low level of physical activity significantly affected mental HR-QoL. Emotional loneliness [β: -8.19; std-β: -3.73; 95%-CI: -4.16 to -3.31] had a stronger association with mental HR-QoL than social loneliness [β: -4.16; std-β: -1.84; 95%-CI: -2.27 to -1.41]. Similar results were observed for the dimensions of loneliness separately. The amount of variance explained by the emotional loneliness model was higher than that by the social loneliness model (32 vs. 24%).’

If the editor desires we could add more interpretation of coefficients to the results-section.

Please see my comments attached. Good luck!

We have incorporated each of the comments of this reviewer:

‘This finding indicates that older citizens who miss an intimate or intense emotional relationship and interventions targeting mental HR-QoL deserve more attention in policy and practice than in the past.’

‘The potentially harmful effects of loneliness on health are well-established.’

‘We characterized loneliness from an emotional and social perspective.’

‘This finding suggest that older citizens who miss an intimate or intense emotional relationship and interventions targeting mental HR-QoL deserve more attention in policy and practice than in the past.’

Reviewer 3 Report

This paper is highly well presented nicely written one.

my minor suggestion is to supplement the tables. for, ex, describe the abbreviated word in full language, and

present the statistical method in use under the tables, and explain the PCS-12 whether the higher score means  better HRQoL.

Living alone proved not to be a risk factor for HRQoL, but loneliness was important risk factor. How can you explain that? would you explain more about that on discussion? 

Author Response

Reviewer 3

This paper is highly well presented nicely written one. My minor suggestion is to supplement the tables. for, ex, describe the abbreviated word in full language, and present the statistical method in use under the tables, and explain the PCS-12 whether the higher score means  better HRQoL.

We also thank this reviewer for his/her compliment. We have now supplemented all the tables (including those in the supplement) by describing the abbreviated words in full language, and by presenting the statistical method in use under the tables. Also we included under each table that higher PCS-12 scores indicate higher levels of health.

Living alone proved not to be a risk factor for HRQoL, but loneliness was important risk factor. How can you explain that? would you explain more about that on discussion?

We have elaborated more on the difference between loneliness and social isolation in the introduction: ‘Because of their decreasing social circle and increasing health restrictions, older citizens may be particularly vulnerable to feeling lonely [1-3]. Opposed to social isolation which is an objective measure of the number of social interactions that people have, loneliness is a subjective measure [4-6]. Loneliness has been described as the unpleasant subjective feeling that occurs when a person’s  social environment is deficient in some important way, either quantitatively or qualitatively [3, 6]. Hence, someone may be bodily around people but still feel lonely or be fully socially isolated but not feel lonely. Social isolation may evoke loneliness and vice versa; and both concepts can arise simultaneously [4-6]. From a policy perspective, alleviating loneliness cannot be achieved by solely increasing the number of social interactions, but by improving meaningful social networks [3, 5].’

In addition, we included in the discussion:

‘[…] loneliness has a stronger association with mental HR-QoL than with physical HR-QoL. […] In addition, living alone was a risk factor for mental (and not physical) HR-QoL only in addition to emotional (and not social) loneliness. These results may imply that citizens who miss an intimate or intense emotional relationship and live alone are at particular risk for a lower mental HR-QoL and confirms that both concepts can arise separately and simultaneously. Earlier studies have already suggested that having a partner and/or having (adult) children relate positively to well-being and mental health [31, 32]. Our findings underline the need to address the two loneliness dimensions separately.’

Round 2

Reviewer 1 Report

The authors have substantially revised the manuscript according to previous comments.